# Isolation, Identification and Antibacterial Mechanism of the Main Antibacterial Component from Pickled and Dried Mustard (*Brassica juncea* Coss. var. *foliosa* Bailey)

**DOI:** 10.3390/molecules27082418

**Published:** 2022-04-08

**Authors:** Shirong Huang, Xiaojie Chen, Rui Yan, Meng Huang, Dongfang Chen

**Affiliations:** Department of Biological and Food Engineering, Xiangtan University, Xiangtan 411105, China; cxiaojie0501@163.com (X.C.); yanrui1115@163.com (R.Y.); 13873237032@163.com (M.H.); cdfcdf0922@163.com (D.C.)

**Keywords:** pickled and dried mustard, antibacterial component, separation, identification, antibacterial activity, antibacterial mechanism

## Abstract

Our previous study showed that the ethyl acetate fraction (EAF) from an ethanolic extract of pickled and dried mustard (*Brassica juncea* Coss. var. *foliosa* Bailey) had significant antibacterial activity. Here, the EAF was further separated into seven sub-fractions by silica gel column chromatography. The antibacterial activities of the EAF and its sub-fractions against *Staphylococcus aureus* and *Pseudomonas fluorescens* were assessed using the agar diffusion method and double dilution method. Among the seven sub-fractions, the third sub-fraction (Fr 3) possessed the strongest antibacterial activity. The main component in Fr 3 was identified by GC-MS, UV-vis, FT-IR, HPLC, ^1^H NMR and ^13^C NMR techniques, and was found to be succinic acid. The content of succinic acid in Fr 3 was determined as 88.68% (*w*/*w*) by HPLC. Finally, the antibacterial mechanism of succinic acid against the tested strains was explored by determining the intracellular component leakage, measuring the cell particle size and observing the cell morphology. The results showed that succinic acid could damage the cell membrane structure and intracellular structure to increase the leakage of cell components and reduce the cell particle size. Our results suggest that succinic acid could be used in food industry to control bacterial contamination by *S. aureus* and *P. fluorescens*.

## 1. Introduction

Microorganism contamination is one of the primary factors related to food spoilage during processing and storage [1]. Food microbial spoilage can not only lead to off-odors and the formation of toxic compounds, gas, and slime, but can also cause enormous financial loss. It is estimated that food loss caused by spoilage microorganisms ranges from 10% to 50% in developing countries, depending on the type of food [2]. Food preservatives are usually added into food stuffs to delay food spoilage and prolong the shelf life. Chemical preservatives are widely used in food processing because of their low cost, broad antimicrobial spectrum, and high efficiency. Since many chemical preservatives are harmful to human health and various harmful food bacteria are resistant to chemical preservatives [3] there is an urgent need to look for effective and safe alternatives to chemical preservatives. Natural preservatives from plants have attracted researchers’ attention because of their rich sources and high safety.

Pickled and dried mustard (PDM, called Meigancai in Chinese) is a traditional fermented vegetable product widely consumed in China. It is shelf stable and can be kept for up to 2 years at room temperature. Traditionally, PDM is obtained by pickling fresh mustard with salt (3–10%, *w*/*w*) for several weeks and then drying. It is commonly used as an ingredient for steamed meat in China. Steamed pork with PDM is a traditional Chinese dish with a delicious taste and flavor that enhances peoples’ appetite. Steamed pork with PDM can be kept at room temperature for more than half a year [4], much longer than steamed pork without PDM. These properties may be mainly attributed to the antimicrobial and antioxidant activities of the PDM.

It has been demonstrated that PDM has strong antioxidant activity [5,6]. However, there is little research on its antimicrobial activities, especially on its antibacterial activities against foodborne spoilage and pathogenic bacteria. Previously, we have evaluated the antimicrobial activities of crude ethanol extract and its various fractions (petroleum ether fraction, ethyl acetate fraction, and the residual fraction) from pickled and dried mustard [7]. The results showed that all samples exhibited potent antibacterial activities against the tested bacteria, among which ethyl acetate fraction (EAF) exhibited the strongest antibacterial activity. Although the antimicrobial properties of the EAF of an ethanol extract from pickled and dried mustard were studied, no research was reported on the determination of the compounds responsible for the antimicrobial activity in the EAF. On the other hand, there are several mechanisms of action among antimicrobial agents, such as the inhibition of protein synthesis, inhibition of metabolic pathways, interference with cell-wall synthesis, inhibition of DNA and RNA synthesis, and lysis of the bacterial membrane [8]. Different mechanisms of action are involved in the antimicrobial activity of antimicrobial agents, due to the variability of quantity and the chemical profiles of the antimicrobial components [9]. There are no reports on the mechanism of action of the main antibacterial components from pickled and dried mustard on the growth of microorganisms.

Based on our previous work, this present study aimed to: (a) use silica gel column chromatography to further separate the ethyl acetate fraction; (b) assess the antibacterial activities of the ethyl acetate fraction and its sub-fractions against *Staphylococcus aureus* and *Pseudomonas fluorescence* to find the most effective component; (c) chemically and structurally characterize the most effective component present in the most effective sub-fraction; (d) explore the antibacterial mode of action of the most effective component against *S. aureus* and *P. fluorescence*.

## 2. Results and Discussion

### 2.1. Antibacterial Activities of EAF and Its Sub-Fractions

The antibacterial activities of the EAF and its sub-fractions were assessed by determining the values of minimum inhibitory concentration (MIC), minimum bactericidal concentration (MBC), and the diameter of the inhibition zone (DIZ).

#### 2.1.1. Determination of Inhibition Zone

Firstly, the antibacterial activities of the EAF against the two tested bacteria were evaluated at concentrations of 10–30 mg/mL. Figure 1 shows the inhibition zones of the EAF against *S. aureus* and *P. fluorescens* at a concentration of 20 mg/mL. Table 1 shows the DIZ values. The EAF showed strong inhibitory effects against *S. aureus* and *P. fluorescens*. Their inhibitory effects were significantly enhanced with the increase in sample concentrations. The DIZ value of the EAF against *S. aureus* was 24.94 mm at the concentration of 30 mg/mL, which was significantly lower than that of tetracycline at 70 μg/mL. The DIZ value against *P. fluorescens* was 18.66 mm at the concentration of 30 mg/mL and significantly higher than that of tetracycline at 70 μg/mL. The inhibition zone diameters of the EAF against *S. aureus* were larger than those of *P. fluorescens*.

The above results demonstrate that the EAF was active against *S. aureus* and *P. fluorescens*; therefore, the fraction was further isolated for identification of the most effective component. The fraction was separated into seven sub-fractions by silica gel column chromatography. The inhibition zones of the sub-fractions against *S. aureus* and *P. fluorescens* at a concentration of 20 mg/mL was also shown in Figure 1. Table 1 shows the DIZ values of the sub-fractions. All of the sub-fractions showed inhibitory effects against both tested bacterial strains, and their antibacterial activities were significantly different. Among the seven sub-fractions, Fr 2 and Fr 3 showed stronger antibacterial activities than the EAF, while Fr 4–Fr 7 exhibited weaker activities. The inhibition activity of Fr 1 against *P. fluorescens* was significantly stronger than that of the EAF, but its activity against *S. aureus* was similar to that of the EAF. Fr 3 showed the highest inhibitory activities against both bacterial strains with inhibition zone diameters of 27.18 mm for *S. aureus* and 18.98 mm for *P. fluorescens,* whereas Fr 7 exhibited the lowest activities. These results indicate that the components with strong antibacterial activity against *S. aureus* and *P. fluorescens* in pickled and dried mustard are of moderate polarity.

#### 2.1.2. Determination of Minimum Inhibitory Concentration (MIC) and Minimum Bactericidal Concentration (MBC)

Table 2 shows the MIC and MBC of the EAF from pickled and dried mustard against the two tested bacteria. The results also showed that the EAF exhibited strong antibacterial activities against *S. aureus* and *P. fluorescens*. The MIC and MBC against *S. aureus* were lower (0.63 and 1.25 mg/mL, respectively), which indicated that the EAF had a stronger bacterial inhibition and bactericidal effect against *S. aureus*. This result is consistent with that of the inhibition zone test. However, the MIC and MBC of the EAF were far higher than those of tetracycline, which suggested that the EAF had a weaker antibacterial activity than tetracycline.

The MIC and MBC of the sub-fractions against *S. aureus* and *P. fluorescens* are also shown in Table 2. As can be seen from the table, the MICs and MBCs of the seven sub-fractions against the two test bacteria were different. Among the sub-fractions, Fr 3 had the minimum MIC and MBC values, while Fr 7 had the maximum values. Furthermore, Fr 2 and Fr 3 were more effective against *S. aureus* than against *P. fluorescens*.

In this study, the Gram-positive bacteria *S. aureus* seemed to be more sensitive to the EAF and its sub-fractions from pickled and dried mustard than the Gram-negative bacteria *P. fluorescence*, except for Fr 7, which had similar antibacterial activity against both strains. This result may be caused by the different structure of the outer membrane between the two strains. Gram-positive bacteria (*S. aureus*) have a single layer peptidoglycan outer membrane, while the Gram-negative bacteria (*P. fluorescence*) possesses a thick layer of lipopolysaccharide, which results in *P. fluorescence* being more resistant to the EFA and its sub-fractions. A similar phenomenon was found on *Escherichia coli* and *S. aureus* treated with cinnamon essential oil [10].

### 2.2. Identification of the Main Antibacterial Component

Based on the results of the antibacterial assay, Fr 3 was found to possess the strongest antibacterial activity; therefore, it was selected for component analysis.

#### 2.2.1. GC-MS Spectral Analysis

Fr 3 was analyzed by GC-MS and its spectrum was shown in Figure 2. Table 3 shows its chemical composition as well as the retention times of the compounds. Three compounds were identified from Fr 3 with a dominance of succinic acid (96.7%). Succinic anhydride and mono-methyl succinate represent 1.68 and 1.62%, respectively. They showed a similar percentage of ≥97%.

#### 2.2.2. UV Spectral Analysis

Figure 3A shows the UV spectra of Fr 3. As a contrast, the UV spectra of succinic acid were also presented. It can be seen that Fr 3 had almost the same UV spectra patterns as succinic acid, with a maximum absorbance at 210 nm.

#### 2.2.3. FT-IR Analysis

The FT-IR spectra of Fr 3 and succinic acid are shown in Figure 3B. The results show that Fr 3 and succinic acid had similar absorption peaks within the wavenumber range from 4000 to 500 cm^−1^. The peaks observed at 3412, 2644 and 2538 cm^−1^ corresponded to the O-H stretching vibrations. The peak at 2931 cm^−1^ was assigned to CH stretching vibration of the CH_2_ groups. The frequency 1694.49 cm^−1^ was attributed to C=O stretching. The peaks at 1415 cm^−1^ and 1309 cm^−1^ were assigned to the C-O-H bending in plane and the OH stretching vibration, respectively. The bands located at 1201 and 914 cm^−1^ were ascribed to the C-C asymmetric and symmetric stretching vibrations, respectively. The peaks at 801, 636 and 581 cm^−1^ corresponded to the C-H bending out of plane, COO- bending and COO- rocking, respectively [11,12,13]. The results show that the observed peak values coincided with the previously reported data. All these data demonstrate that Fr 3 is mainly composed of succinic acid.

#### 2.2.4. ^1^H NMR and ^13^C NMR Spectral Analysis

The ^1^H NMR and ^13^C NMR spectra of Fr 3 are shown in Figure 4. The ^1^H NMR spectrum of Fr 3 indicated the presence of two methylene protons at δ_H_ 2.57, while the carbon NMR (^13^C NMR) spectrum indicated signals for two methylene carbons (δ_C_ 28.41), and two carboxylic carbons (δ_C_ 174.76). These NMR data of Fr 3 were highly similar to that of succinic acid [14].

#### 2.2.5. HPLC Chromatogram Analysis

The HPLC chromatogram of the Fr 3 and succinic acid standard is shown in Figure 5. Both the Fr 3 and succinic acid standard showed one major peak at a retention time of 2.8 min, which suggests that the main component of Fr 3 is succinic acid. The standard curve fitting equation of the succinic acid standard is y = 980.61x + 87.20, R^2^ = 0.9994 (the x is the concentration of the succinic acid standard (mg/mL), and the y is the peak area). The succinic acid content in Fr 3 was determined to be 87.88% from the standard curve. Succinic acid is an intermediate product of the tricarboxylic acid cycle and is the end product of microbial fermentation [15]. It is widely found in fermented vegetables. Zhao and Ding reported that the main organic acids are lactic, acetic, citric, succinic, and malic acid in the brine of pickled *Brassia juncea* Coss. var. *multiceps* Tsen et Lee [16]. Zhao and Yang determined the organic acids in the brine of traditionally pickled *Brassica juncea* Coss. via ion chromatography and found that the important organic acid was succinic acid, with a content of 0.82% (*w*/*w*) [17]. However, Ru et al., found that no succinic acid was detected in 18 kinds of preserved mustards from different producing areas in China [18,19]. The differences in succinic acid content among the pickled vegetables may be due to the different raw material and fermentation process used.

#### 2.2.6. Antibacterial Activity of Succinic Acid

After the above analysis, it was found that succinic acid was the main component in Fr 3. To further confirm this, the antibacterial activity of Fr 3 was compared with that of the succinic acid standard at the same concentration. The results showed that succinic acid exhibited strong antibacterial activities against *S. aureus* and *P. fluorescens* with DIZ values of 27.57 mm and 18.90 mm, respectively. The MIC values of succinic acid against the two bacteria were 0.31 and 0.63 mg/mL, and the MBC values were 0.31 and 0.63 mg/mL, respectively. These values are comparable to those of Fr 3, which also demonstrated that Fr 3 was mainly composed of succinic acid.

### 2.3. Effect of Succinic Acid on the Integrity of the Cell Membrane

Cell membrane integrity is a key factor for the survival and growth of bacteria [10]. By measuring the leakage of bacterial cell constituents, we can assess whether the cell membrane is damaged [20]. Figure 6 shows the total nucleotide leakage and total protein leakage from *S. aureus* and *P. fluorescens* treated with succinic acid at different concentrations (0, 1 × MIC, 2 × MIC and 4 × MIC). For 0 h, shortly after treatment with succinic acid, the release of nucleic acids from *S. aureus* treated with succinic acid at 1 × MIC, 2 × MIC and 4 × MIC were 2.12, 2.55, and 2.60 times higher than that of the control, respectively, while the release from *P. fluorescens* increased by 1.88, 2.00, and 1.97 times, respectively. Compared to the control, treatment with different concentrations of succinic acid significantly increased the leakage of nucleic acids from both bacteria (*p* < 0.05). In general, the cell leakage increases with increasing exposure time and concentration. However, no significant differences in the amount of protein leakage were found between the bacteria exposed to succinic acid at a concentration of 1 × MIC and the control group (*p* > 0.05). For 0 h, compared with the control, protein leakage from *S. aureus* treated with succinic acid at 1 × MIC, 2 × MIC and 4 × MIC were 1.18, 1.25, and 1.26 times, respectively, while the leakage from *P. fluorescens* were 1.02, 1.12, and 1.29 times, respectively. It seemed that proteins leaked more slowly than nucleic acids within 2 h after exposure to succinic acid. Similar results were obtained by [21] who found that proteins leaked more slowly than nucleic acids within 2 h from *Aeromonas hydrophila* exposed to dithiocyano-methane.

Our results indicated that the cell membrane integrity of the two bacteria was damaged under the treatment of succinic acid, leading to the leakage of nucleic acids and proteins from the bacterial cells. Similarly, other antibacterial agents, such as *Aronia melanocarpa*, *Chaenomeles superba*, and *Cornus mas* leaf extracts [22], hawthorn methanol extract [23], mandarin essential oil [24], cinnamon essential oil [25], peppermint essential oil [12], and dodecyl gallate microencapsulated with methyl-β-cyclodextrin [26] have also caused the leakage of nucleic acids and proteins in *S. aureus*. Liu, Mei, and Xie [27] reported that daphnetin could damage the *P. fluorescens* cells, resulting in the leakage of proteins and nucleic acids.

### 2.4. Effect of Succinic Acid on Average Size

It was reported that the cell size could be a good indication of the physiological state of the tested strain [28]. Antimicrobial agents could influence the physiological state of the tested strain, resulting in changes in cell size, and the effects were varied within different strains. Figure 7 presents the average size of *S. aureus* and *P. fluorescens* treated with succinic acid at 0, 1× and 2 × MIC after 0, 2 and 24 h. At the start time (0 h), *S. aureus* treated with 2 × MIC succinic acid had a significantly larger average size than the control group (*p* < 0.05), while no significant difference was found between the treatment group with 1 × MIC succinic acid and the control group (*p* > 0.05). However, *P. fluorescens* treated with 1 × MIC or 2 × MIC succinic acid had a significantly lower average size than the control group (*p* < 0.05). After exposure to the succinic acid solution for 2 h and 24 h, the average cell sizes of *S. aureus* and *P. fluorescens* were significantly smaller than those of the control groups (*p* < 0.05). In addition, both bacteria treated with succinic acid for 24 h had significantly smaller average sizes than those treated for 0 h (*p* < 0.05). The results indicated that the average sizes of both bacteria decreased after treatment with succinic acid. Furthermore, during the exposure time (0, 2 and 24 h), *S. aureus* in both the treatment and control groups had significantly smaller average sizes than *P. fluorescens* (*p* < 0.05).

Succinic acid had a great effect on the average sizes of the two strains, but the effect on each strain was different. After exposure to succinic acid for 24 h, the average particle size of *S. aureus* decreased by 40%, while that of *P. fluorescens* decreased by 15%. The study of Wang, Chang, Yang and Cui [28] showed that this mainly depends on the different physiological states of the tested bacteria.

### 2.5. Effect of Succinic Acid on Cell Morphology

The morphological changes induced by 2 × MIC of succinic acid in both bacteria were investigated by TEM and the results were presented in Figure 8. The control *S. aureus* and *P. fluorescens* had normal structures of bacterial cells (Figure 8a,c). The cell membranes are intact and clearly visible, and which completely surround the cell body. The cytoplasm is compact and evenly distributed. *S. aureus* after treatment (Figure 8b) showed an obvious change in characteristics. A cavity occurred in the cells, the cell membrane ruptured, and the cell contents leaked, causing the cell edges to blur. Treated *P. fluorescens* (Figure 8d) also showed significant change. The cell walls and cell membranes were broken. Moreover, the cell contents extravasated, forming many obvious pits and gaps. However, despite the significant leakage of the cell’s constituents from both bacteria exposed to succinic acid, the general morphological structures of the bacterial cells were retained. Based on the TEM images and the leakage of nucleic acids and proteins, succinic acid interacted with the cell membrane, leading to pore formation. Consequently, the intracellular components (e.g., nucleic acids and proteins) were leaked and the cell was inactivated. Similar results were obtained by Wang, Chang, Yang and Cui [28] who observed the morphology changes of *Salmonella Enteritidis*, *Escherichia coli* and *Listeria monocytogenes* before and after treatment with lactic acid by TEM. It was reported that polyphenols [29], Liangguoan [30], quinic acid [31], and vine tea extract [32] could also rupture the cell membrane of *S. aureus* and lead to leakage of the cell contents. Liu, Mei, and Xie [27] found that daphnetin could destroy, lyse, and break the cell walls and cell membranes of *P. fluorescens*, resulting in cytoplasm outflow and cavity formation.

## 3. Materials and Methods

### 3.1. Materials

Pickled and dried mustards (*Brassica juncea* Coss. var. *foliosa* Bailey) were obtained from a rural market in Guangchang County, Jiangxi Province, China. They were obtained by pickling the mustards with 5% salt (*w*/*w*) for 15 d and then drying them in the sun.

The *Staphylococcus aureus* cultures were provided by the Microbiology Laboratory, Xiangtan University, while the *Pseudomonas fluorescens* (strain no. 21620) was purchased from the China Center of Industrial Culture Collection (CICC).

### 3.2. Preparation and Fractionation of Ethanol Extract

The ethanol extract of PDM was prepared by following a previously reported method [7], with slight modification. Briefly, the pickled and dried mustard was dried and ground to obtain a uniform powder (screened through a 40-mesh sieve). The powder (4.0 kg) was extracted at room temperature for 48 h with anhydrous ethanol (20 L). The mixture was filtered under vacuum and the residue was extracted for one more time according to the above conditions. After filtration, the filtrates were combined and concentrated under reduced pressure at 45 °C using a RE-2000A rotary vacuum evaporator (Yarong Biochemical Equipment & Instrument Co., Ltd., Shanghai, China) to obtain 431.76 g of ethanol extract. Then, the extract was suspended in distilled water, and successively partitioned with petroleum ether (boiling range 30–60 °C, 5 times) and then ethyl acetate (5 times). The ethyl acetate phase was concentrated under reduced pressure to yield the ethyl acetate fraction for isolation of the desired compounds.

### 3.3. Isolation of the Main Antibacterial Compounds

The EAF was separated by silica gel column chromatography (40 cm × 5.5 cm in size). Four grams of the EAF was filled on top of the column packed with silica gel (200~300 mesh). The column was eluted by using various proportions of petroleum ether, chloroform, and methanol. First, the elution was performed with a mixture of petroleum ether and ethyl acetate mixed in ratios of 3:1, 1:1, and 1:3 followed by ethyl acetate: methanol mixed in the ratios of 10:0, 9:1, 7:3, 5:5, and 0:10. Forty fractions of 100 mL each were collected, and then analyzed by thin layer chromatography (TLC). Fractions with similar TLC profiles were combined to yield six sub-fractions (Fr 1, A2, Fr 4, Fr 5, Fr 6, and Fr 7) as follows: Fr 1 (1–11, 0.107 g); A2 (12–19); Fr 4 (20–26, 0.2580 g); Fr 5 (27–32, 0.4017 g); Fr 6 (33–37, 0.1435 g); Fr 7 (38–40, 0.0811 g). It was found that a white crystal precipitated from the sub-fraction A2 during standing at room temperature. After filtration and recrystallization using ethyl acetate, a white crystal (Fr 3, 0.4271 g) was obtained. The obtained filtrate was evaporated under reduced pressure to yield a yellow oily substance (Fr 2, 1.9860 g). The other sub-fractions were subjected to complete dryness.

### 3.4. Antibacterial Assays

The antibacterial activities of the EAF and its sub-fractions from pickled and dried mustard were investigated. Bacterial strains, such as *S. aureus* and *P. fluorescence* were used in the antibacterial assay.

#### 3.4.1. Activation of Bacteria

*S. aureus* and *P. fluorescence* were inoculated at 37 °C and 30 °C for 24 h in a nutrient broth (NB) for activation, respectively. The activated bacteria were inoculated into the culture medium. After being incubated for another 18 h, the cultured cells were diluted with the corresponding liquid medium to a density of 1 × 10^8^ CFU/mL.

#### 3.4.2. Determination of Antibacterial Activity

The antibacterial activities of the EAF and its sub-fractions were determined using an agar diffusion method with some modifications [33,34]. Firstly, the liquid medium containing 1.5% (*w*/*w*) agar (solid medium) was poured into the Petri dish. After the medium was solidified, sterilized Oxford cups (7.8 mm in diameter) were placed on it. Then, the solid medium containing 1% (*v*/*v*) of each bacterial suspension (10^8^ CFU/mL) was added into the culture dish at the same volume and fully mixed. After the medium had solidified, the Oxford cups were removed, and round wells were formed. Then, 70 µL of various concentrations of the EAF (10–30 mg/mL) or 20 mg/mL of the sub-fraction solutions were poured into the round wells, respectively. Equal amounts of ethanol solution were used as a negative control and 70 µg/mL tetracycline was used as a positive control. The plates were kept for 2 h at 4 °C before being incubated for 24 h at 37 °C for *S. aureus* or at 30 °C for *P. fluorescence*. After incubation, the inhibition zones around the wells were recorded to determine the antibacterial activity.

#### 3.4.3. Determination of Minimum Inhibitory Concentration (MIC) and Minimum Bactericidal Concentration (MBC)

The MIC and MBC of the EAF and its sub-fractions were determined by serial dilution, as described by Lee [35,36]. Stock solutions of the EAF and its sub-fractions were prepared by dissolving them in absolute ethyl alcohol to a final concentration of 400 mg/mL. Two-fold serial dilutions from the stock solution were made ranging from 200 mg/mL to 3.125 mg/mL using ethanol. Then, a 0.1 mL series of concentrations of the EAF and its sub-fractions were mixed with 1.8 mL melted NBA (NB + 1.0% agar) contained in sterile test tubes. Finally, 0.1 mL bacterial suspensions were added into the test tubes and mixed fully, making the final concentrations of the samples between 0.1563–20 mg/mL. Then, the mixtures were poured into sterilized Petri dishes and incubated at the optimum temperature of the tested bacteria (*S. aureus* at 37 °C, *P. fluorescence* at 30 °C) for 24 h. Each experiment was performed in triplicate. The MIC was defined as the minimum concentration of the samples inhibiting microbial growth. The culture was incubated for a further 24 h. The MBC was reported as the minimum concentration at which there was no visible growth. Equal amounts of ethanol solution and 70 µg/mL tetracycline were used as a negative control and positive control, respectively.

### 3.5. Identification of the Main Antibacterial Components

Since the third sub-fraction was found to possess the strongest antibacterial activity, its composition was analyzed to find out the main antibacterial components.

The isolated compound was first analyzed in the Shimadzu GCMS-QP2010 Plus. The oven temperature was set at 40 °C and held for 1 min, then it was raised to 250 °C at a rate of 15 °C/min with a 2 min hold. The temperature of the ions source was set at 200 °C, with an interface at 250 °C. The carrier gas was helium, and its flow rate was 1 mL/min. The injection volume was 0.2 μL and the scan range was 30–500 amu. The identification of the isolated compounds was made by comparing the experimental data with the MS reference database of NIST software (National Institute of Standards and Technology, Gaithersburg, MD, USA).

The UV-vis spectra were measured in methanol with an Agilent CARY60 UV-Vis spectrophotometer. The infrared spectra were reported by a Nicolet 380 FT-IR spectrometer (Thermo Nicolet, MA, USA). The NMR spectra were measured by an AVANCE Ⅲ HD 400 NMR spectrometer (Bruker Biospin, Rheinstetten, Germany) at 500 MHz (^1^H NMR) and 125 MHz (^13^C NMR), respectively, with CD_3_OD as the solvent and tetramethylsilane as the internal standard.

The composition of Fr 3 was also analyzed by using an HPLC system (Agilent Technologies, 1260 Infinity, CA, USA) equipped with a diode array UV detector. The chromatographic column was ODS-C18 (4.6 mm × 250 mm, 5 µm), and the column temperature was 25 °C. The mobile phase consisted of methanol and water in a volume ratio of 60:40 with a flow rate of 1 mL/min. The sample with a concentration of 20 mg/mL was prepared with an HPLC grade methanol and was filtered (0.22 µm microporous membrane). A 20-µL aliquot of the sample was injected and the effluent was monitored at 210 nm. Identification was made by comparing it with the retention time of the succinic acid standard, and the succinic acid content was calculated according to the calibration curves of the standard and the peak area.

### 3.6. Antibacterial Mechanism

#### 3.6.1. Integrity of Cell Membrane

The cell membrane integrity was determined by measuring the release of the cell constituents, including protein and nucleic acids, into the supernatant according to the method described by [12] and [37] with some modifications. *S. aureus* and *P. fluorescens* were cultured in NB at 37 °C and 30 °C for 12–14 h, respectively. After incubation, the suspensions of the tested bacteria were centrifuged at 5000 r/min for 10 min. The cells were collected, washed 3 times with a 0.1 mol/L phosphate buffer solution (PBS, pH 7.4), and resuspended in the same buffer. The washed suspensions were incubated for 2, 6, and 24 h in the presence of succinic acid at the final concentration of 1 × MIC, 2 × MIC or 4 × MIC. After the suspensions were centrifuged at 8000 r/min for 5 min, the supernatants were collected, diluted with PBS, and the absorbance at 260 nm was measured.

An untreated sample was used as a control. The amount of nucleic acid leakage was determined by the absorbance at 260 nm.

In addition, the supernatant was also used to determine the amount of protein leakage. The protein concentration in the supernatant was determined according to the Bradford’s method [38]. Firstly, 1 mL Coomassie brilliant blue G-250 was added into 0.1 mL supernatant and mixed well. After the mixture was incubated in the dark for 2 min, the absorbance at 590 nm was measured. The protein concentration was calculated from a calibration curve prepared with bovine serum albumin.

#### 3.6.2. Size Analysis

Size analysis was performed according to the method of [28]. Bacterial suspensions of *S. aureus* and *P. fluorescens* treated with 1 × MIC and 2 × MIC of succinic acid were obtained as described in Section 3.6.1. The average particle size diameter of the two bacterial cell suspensions was determined by using a MASTERSIZER 2000 laser particle size analyzer after being cultured for 0, 2 and 24 h.

#### 3.6.3. Transmission Electron Microscopy (TEM) Observation

A TEM observation was carried out according to the methods of [28] and [39], with slight modification. *S. aureus* and *P. fluorescens* were cultured in NB at 37 °C and 30 °C for 12–14 h, respectively; thereby, obtaining their bacterial suspensions. Then, succinic acid was added to reach the final concentration of 2 × MIC and the obtained bacterial suspensions were incubated at their optimum growth temperature for 4 h. After incubation, the suspensions were centrifuged for 10 min at 2000 r/min, and the cells were collected and washed 3 times with a 0.1 mol/L of PBS (pH 7.4). The washed cells were then fixed with 2.5% (*w*/*v*) glutaraldehyde at 4 °C overnight. The treated cells were collected by centrifugation. After being washed 3 times with the PBS, the cells were postfixed for 1–2 h with 1% OsO_4_. The mixture was centrifuged, and the cell pellet was washed 3 times with the PBS. Then, the samples were prepared by dehydration with different concentrations of acetone (50%, 70%, 90%, 100%), embedding, ultrathin section, and staining with 3% uranium acetate and lead nitrate. The prepared samples were observed and photographed with a transmission electron microscopy (HT7700, Hitachi, Japan).

### 3.7. Data Analysis

The statistical analyses were performed with SPSS 19.0 software (IBM, Armonk, NY, USA) and the results were expressed as mean values ± standard deviation (SD). The statistical differences between the samples were analyzed by one-way analysis of variance (ANOVA) and Duncan’s multiple comparison tests.

## 4. Conclusions

This study reported the isolation, identification, and antibacterial mechanism of the most effective antibacterial component from pickled and dried mustard (*Brassica juncea* Coss. var. *foliosa* Bailey) against *S. aureus* and *P. fluorescens*. The ethyl acetate fraction from an ethanolic extract of the PDM showed significant antibacterial activity against the two tested strains (*p* < 0.05). Separation of the ethyl acetate fraction led to a sub-fraction possessing the strongest antibacterial activity. As the major component of the sub-fraction, succinic acid is responsible for the antibacterial activities. Succinic acid damaged the cell membranes structures of both bacteria, which led to the leakage of nucleic acids and proteins, and a decrease in cell particle size. The results indicated that the antibacterial substances with better activity could be isolated from the pickled and dried mustard, and the PDM could be used as a potential source of natural antibacterial agent for food preservation.

## Figures and Tables

**Figure 1 molecules-27-02418-f001:**
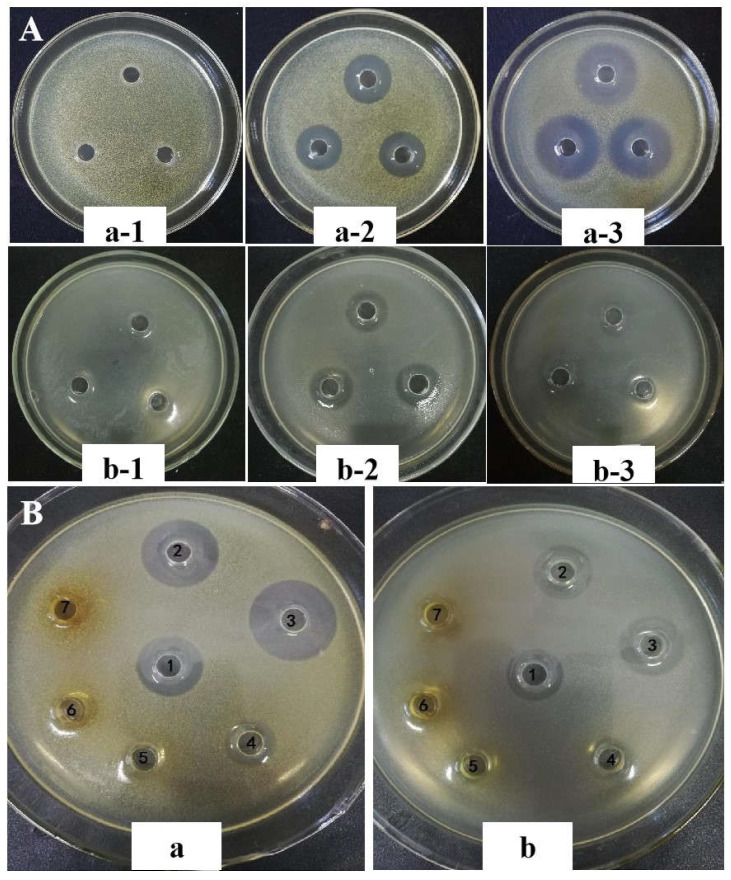
Inhibition zones of the ethyl acetate fraction (**A**) and its sub-fractions (**B**) from pickled and dried mustard against *S. aureus* (a) and *P. fluorescens* (b). (a-1) *S. aureus* control (treated with ethanol); (a-2) *S. aureus* treated with 20 mg/mL ethyl acetate fraction; (a-3) *S. aureus* treated with 70 μg/mL tetracycline; (b-1) *P. fluorescens* control (treated with ethanol); (b-2) *P. fluorescens* treated with 20 mg/mL ethyl acetate fraction; (b-3) *P. fluorescens* treated with 70 μg/mL tetracycline. No. 1 to No. 7 were Fr 1–Fr 7.

**Figure 2 molecules-27-02418-f002:**
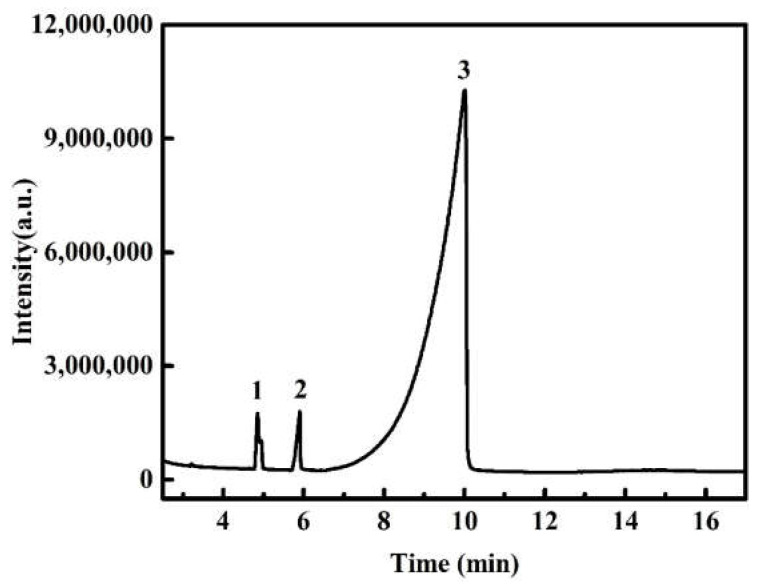
GC-MS spectra of Fr 3; 1: succinic anhydride; 2: mono-methyl succinate; 3: succinic acid.

**Figure 3 molecules-27-02418-f003:**
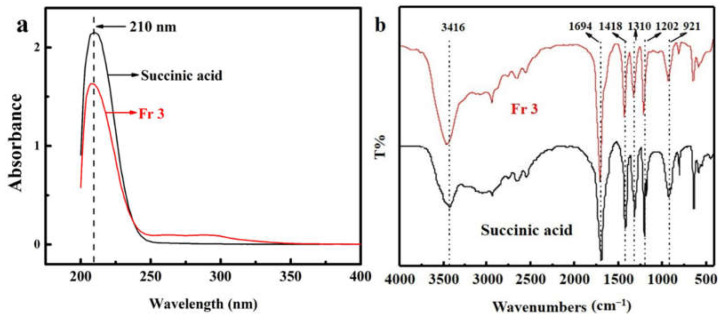
UV-vis absorbance spectra (**a**) and FT-IR spectra (**b**) of Fr 3 and succinic acid.

**Figure 4 molecules-27-02418-f004:**
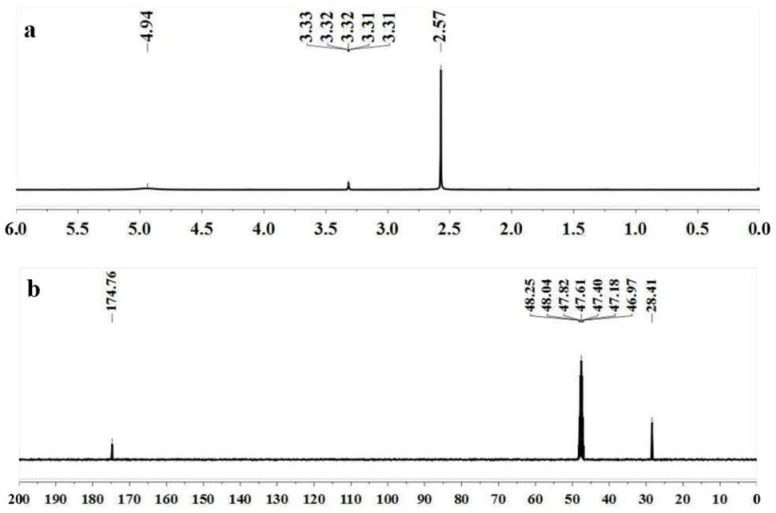
^1^H NMR (**a**) and ^13^C NMR (**b**) spectra of Fr 3.

**Figure 5 molecules-27-02418-f005:**
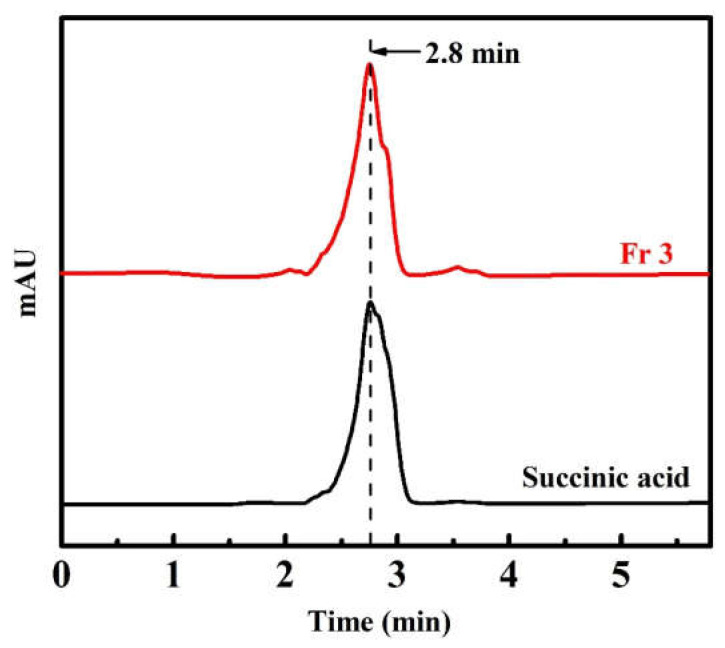
HPLC chromatogram of the Fr 3 and succinic acid standard.

**Figure 6 molecules-27-02418-f006:**
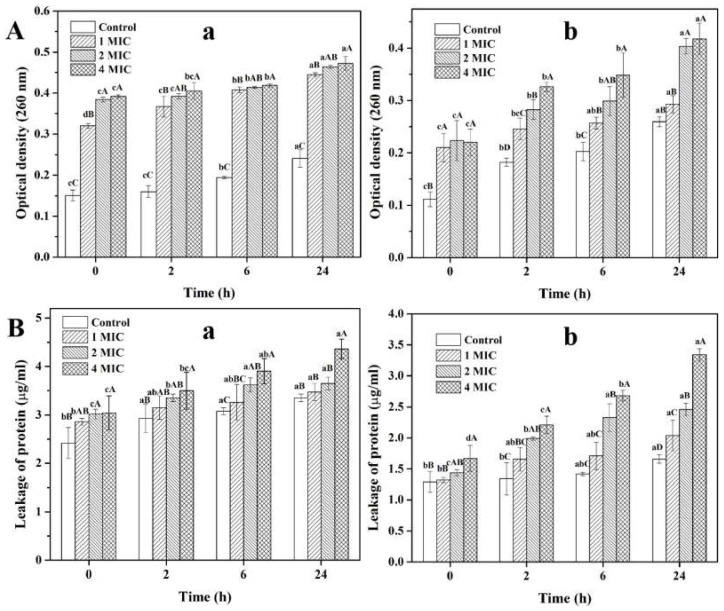
Total nucleotide leakage (**A**) and total protein leakage (**B**) from *S. aureus* (a) and *P. fluorescens* (b) treated by succinic acid. Different lowercase letters indicate significant difference among treatments at different exposure time (*p* < 0.05), while different capital letters indicate significant difference among treatments at different concentrations of succinic acid (*p* < 0.05).

**Figure 7 molecules-27-02418-f007:**
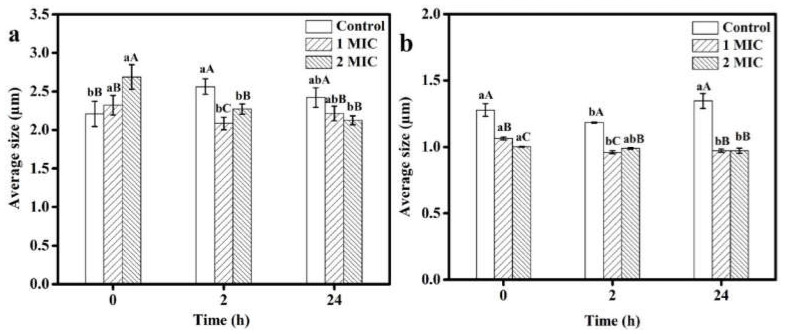
Average sizes of *S. aureus* (**a**) and *P. fluorescens* (**b**) cells treated with succinic acid after 0 h, 2 h and 24 h. Different lowercase letters indicate significant difference among treatments at different exposure times (*p* < 0.05), while different capital letters indicate significant difference among treatments at different concentrations of succinic acid (*p* < 0.05).

**Figure 8 molecules-27-02418-f008:**
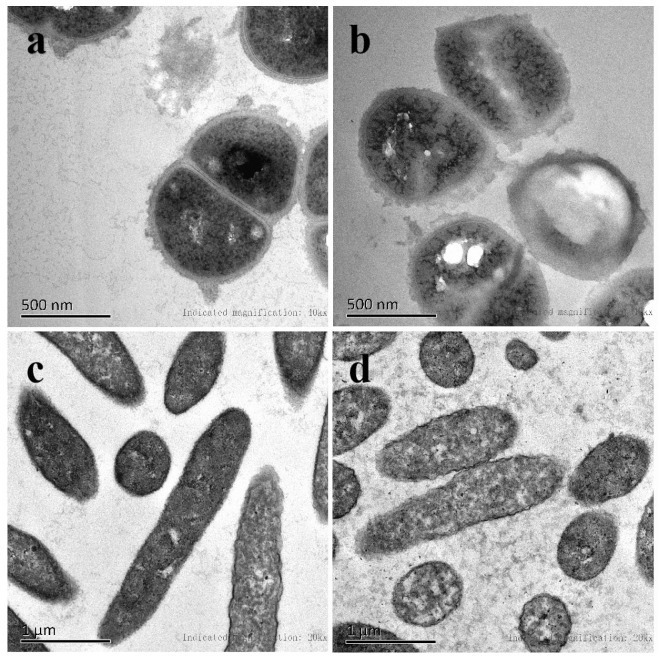
TEM images of *S. aureus* and *P. fluorescens*: (**a**) untreated *S. aureus*; (**b**) *S. aureus* treated with 2 × MIC succinic acid; (**c**) untreated *P. fluorescens*; and (**d**) *P. fluorescens* treated with 2 × MIC succinic acid.

**Table 1 molecules-27-02418-t001:** Diameters of inhibition zones of ethyl acetate fraction and its sub-fractions from pickled and dried mustard.

Samples	Concentration (mg/mL)	Diameter of Inhibition Zone (mm)	
		*Staphylococcus aureus*	*Pseudomonas fluorescens*
Ethyl acetate fraction	10	13.55 ± 0.57f	11.19 ± 0.18f
	20	20.02 ± 0.81d	15.16 ± 0.34d
	30	24.84 ± 0.70b	18.66 ± 0.39a
Fr 1	20	20.07 ± 0.36d	17.51 ± 0.25c
Fr 2	20	22.02 ± 0.52c	18.13 ± 0.13b
Fr 3	20	27.18 ± 0.16a	18.98 ± 0.62a
Fr 4	20	14.61 ± 0.25e	12.15 ± 0.32e
Fr 5	20	11.73 ± 0.50g	10.82 ± 0.35f
Fr 6	20	11.03 ± 0.30h	10.15 ± 0.55g
Fr 7	20	8.44 ± 0.39i	9.35 ± 0.49h
Tetracycline	70 ^#^	27.47 ± 0.80a	15.18 ± 0.60d

^#^ The unit is μg/mL. Different lowercase letters in the same column represent significant difference (*p* < 0.05).

**Table 2 molecules-27-02418-t002:** MIC and MBC of the ethyl acetate fraction and its sub-fractions from pickled and dried mustard.

Samples	*Staphylococcus aureus*		*Pseudomonas fluorescens*	
	MIC	MBC	MIC	MBC
Ethyl acetate fraction	0.63	1.25	1.25	2.50
Fr 1	0.63	1.25	1.25	1.25
Fr 2	0.63	0.63	1.25	1.25
Fr 3	0.31	0.31	0.63	0.63
Fr 4	1.25	2.50	2.50	2.50
Fr 5	2.50	5.00	5.00	5.00
Fr 6	2.50	5.00	5.00	5.00
Fr 7	5.00	10.00	5.00	10.00
Tetracycline	0.63 × 10^−3^	0.63 × 10^−3^	2.50 × 10^−3^	2.50 × 10^−3^

Note: Data are expressed in mg/mL. MIC: minimum inhibitory concentration; MBC: minimum bactericidal concentration.

**Table 3 molecules-27-02418-t003:** Identified components from Fr 3 by GC-MS analysis.

Peak No.	Retention Time (min)	Chemical Formula	Name of the Compounds	Peak Area (%)	Match (%)
1	4.854	C_4_H_4_O_3_	Succinic anhydride	1.68	99
2	5.900	C_5_H_8_O_4_	Mono-methyl succinate	1.62	97
3	10.010	C_4_H_6_O_4_	Succinic acid	96.7	98

## Data Availability

Not applicable.

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
