# Peer review of "Isolation, Identification and Antibacterial Mechanism of the Main Antibacterial Component from Pickled and Dried Mustard (Brassica juncea Coss. var. foliosa Bailey)"

_molecules, 2022, doi:10.3390/molecules27082418_

Round 1
Reviewer 1 Report
The authors have isolated, identified and determined the mode of antibacterial action for the most abundant component in pickled and dried mustard. There are some minor corrections that should be taken care.
Lines 59-60,63 please write bacterial names in italic
Line 67 write full name for MIC and MBC first time it is mentioned
Line 105 please change the title of this subsection
Line 188 write bacteria italic
207-208, 211, 212 write species name in italic
Line 214 why did you study the cell size? Please include more references here on the importance on obtaining such data
Lines 269-271 do these strains have some numbers/codes
Author Response
Dear Editor and Reviewers:
Thank you for your quick response and valuable suggestions. We have revised the manuscript accordingly. The revisions we have made are as follows. Please check it.
The comment: Lines 59-60,63 please write bacterial names in italic. Line 67 write full name for MIC and MBC first time it is mentioned. Line 188 write bacteria italic. 207-208, 211, 212 write species name in italic.
Response: Thank you for pointing out our typo. We have corrected the mistakes.
The comment: Line 105 please change the title of this subsection.
Response: We have changed the title to “Determination of minimum inhibitory concentration (MIC) and minimum bactericidal concentration (MBC)”.
The comment: Line 214 why did you study the cell size? Please include more references here on the importance on obtaining such data.
Response: It was reported that the cell size could be a good indication of the physiological state of the tested strain. Antimicrobial agent could influence the cell size greatly, and it had different effects on different strains. Our results showed that succinic acid had a great effect on the average sizes of the two tested strains, and the effect on each strain was different. The results suggested that the antibacterial effect of succinic acid was probably caused by the physiological changes of the bacterial cells.
We have added the following paragraph into the revised manuscript (Line 245-248).
“It was reported that the cell size could be a good indication of the physiological state of the tested strain. Antimicrobial agent could influence the physiological state of the tested strain, resulting in changes in cell size, and the effects were varied within different strains.”
The comment: Lines 269-271 do these strains have some numbers/codes.
Response: The Staphylococcus aureus strain has no numbers. The Pseudomonas fluorescens strain was purchased from China Center of Industrial Culture Collection (CICC) and its number was 21620. We have added this in the revised manuscript.
Reviewer 2 Report
The article is interesting and well written. I have only few little suggestions.
In the introduction, please add a description of antibacterial mechanism of action of extracts.
Line 59 and 60 and 63, 212 – the names of bacteria should be italic
Line 74, 76, 176 – in my opinion it is impossible to measure the inhibition zone so accurately? Please explain how you did it or change the results. For instance, change 24.94 mm to 25 mm. Check the whole manuscript.
Paragraph 2.1 – There is a lack of discussion. Please discuss the results with literature data. Was antibacterial activity of pickled and dried mustard evaluated in the literature? You showed that Gram-positive bacteria S. aureus was more sensitive than Gram-negative P. fluorescens. Can you explain why? Is it in consistent with literature data?
Paragraph 2.2 - According to literature, is succinic acid usually dominant in mustard? Discuss it with literature data.
Table 2 – please standardize an order of magnitude. MIC and MBC are 0.625; 1.25; 0.3125. In addition it is not necessary to give 4 decimals (0.3125). Usually two decimals are enough. Check it in the whole manuscript.
Lin 208 – the names of plants itallic
Author Response
Dear Editor and Reviewers:
Thank you for your quick response and valuable suggestions. We have revised the manuscript accordingly. The revisions we have made are as follows. Please check it.
The comment: In the introduction, please add a description of antibacterial mechanism of action of extracts.
Response: Thank you for your valuable suggestions. The following paragraph was added in the revised manuscript (Line 56-65).
“Although the antimicrobial properties of the EAF of ethanol extract from pickled and dried mustard were studied, no researches were reported on the determination of the compounds responsible for the antimicrobial activity in the EAF. On the other hand, there are several mechanisms of action among antimicrobial agents, such as inhibition of protein synthesis, inhibition of metabolic pathways, interference with cell-wall synthesis, inhibition of
DNA and RNA synthesis, and lysis of the bacterial membrane. Different mechanisms of action are involved in the antimicrobial activity of antimicrobial agents, due to the variability of quantity and chemical profiles of the antimicrobial components. There are no reports on the mechanism of action of the main antibacterial components from pickled and dried mustard on the growth of microorganisms.”
The comment: Line 59 and 60 and 63, 212 – the names of bacteria should be italic. Lin 208 – the names of plants italic.
Response: Thank you for pointing out our typo. We have corrected the mistakes.
The comment: Line 74, 76, 176 – in my opinion it is impossible to measure the inhibition zone so accurately? Please explain how you did it or change the results. For instance, change 24.94 mm to 25 mm. Check the whole manuscript.
Response: We have determined the inhibition zone by using the agar diffusion method. Firstly, solid medium was poured into the Petri dish when it was hot. After the medium was solidified, sterilized Oxford cups were placed on it. Then the bacterial suspension was added into the culture dish and spread. The Oxford cups were removed and round wells were formed. Then the antibacterial solutions were poured into the round wells. The plates were incubated for 24 h at 37°C for S. aureus or at 30°C for P. fluorescence. After incubation, the diameter of the clear zone around the well was measured in two cross directions using a vernier caliper and the average values were recorded. The experiment was repeated three times and results were expressed as mean values ± standard deviation.
The comment: Paragraph 2.1 – There is a lack of discussion. Please discuss the results with literature data. Was antibacterial activity of pickled and dried mustard evaluated in the literature? You showed that Gram-positive bacteria S. aureus was more sensitive than Gram-negative P. fluorescens. Can you explain why? Is it in consistent with literature data?
Response: Thank you for your good comments. At present, we have not found any literature related to the antibacterial activity of the ethyl acetate fraction and its sub-fractions from pickled and dried mustard. The reasons that S. aureus was more sensitive to the EAF and its sub-fractions from pickled and dried mustard than P. fluorescens are as follows.
“In this study, the gram-positive bacteria S. aureus seemed to be more sensitive to the EAF and its sub-fractions from pickled and dried mustard than the gram-negative bacteria P. fluorescence, except for Fr 7, which had similar antibacterial activity against both strains. This result may be caused by the different structure of outer membrane between the two strains. Gram-positive bacteria (S. aureus) have a single layer of peptidoglycan outer membrane, while the gram-negative bacteria (P. fluorescence) possess a thick layer of lipopolysaccharide which results in P. fluorescence more resistant to the EFA and its sub-fractions. Similar phenomenon was found on Escherichia coli and Staphylococcus aureus treated with cinnamon essential oil.”
We have added the above paragraph to the revised manuscript (Line 134-142).
The comment: Paragraph 2.2 - According to literature, is succinic acid usually dominant in mustard? Discuss it with literature data.
Response: Thank you for your valuable suggestions. We have added the following discussions into the revised manuscript (Line 188-198).
“Succinic acid is an intermediate product of the tricarboxylic acid cycle and is the end product of microbial fermentation. It is widely found in fermented vegetables. Zhao and Ding reported that the main organic acids was lactic, acetic, citric, succinic and malic acid in the brine of the pickled Brassia juncea Coss. var. multiceps Tsen et Lee. Zhao and Yang determined the organic acids in the brine of traditionally pickled Brassica juncea Coss. by ion chromatography and found that the important organic acid was succinic acid, with a content of 0.82% (w/w). However, Ru et al. found that no succinic acid was detected in 18 kinds of preserved mustards from different producing areas in China. The differences in succinic acid content among the pickled vegetables may be due to the different raw material and fermentation process used.”
The comment: Table 2 – please standardize an order of magnitude. MIC and MBC are 0.625; 1.25; 0.3125. In addition it is not necessary to give 4 decimals (0.3125). Usually two decimals are enough. Check it in the whole manuscript.
Response: Thank you for your valuable comments. We have normalized the order of magnitude to two decimals as suggested.